# REGULARLY VARYING REPRESENTATION FOR SENTENCE EMBEDDING

## ABSTRACT

The dominant approaches to sentence representation in natural language rely on learning embeddings on massive corpuses. The obtained embeddings have desirable properties such as compositionality and distance preservation (sentences with similar meanings have similar representations). In this paper, we develop a novel method for learning an embedding enjoying a dilation invariance property. We propose two algorithms: *Orthrus*, a classification algorithm, constrains the distribution of the embedded variable to be regularly varying, *i.e.* multivariate heavy-tail. and uses Extreme Value Theory (EVT) to tackle the classification task on two separate regions: the tail and the bulk. *Hydra*, a text generation algorithm for dataset augmentation, leverages the invariance property of the embedding learnt by *Orthrus* to generate coherent sentences with controllable attribute, *e.g.* positive or negative sentiment. Numerical experiments on synthetic and real text data demonstrate the relevance of the proposed framework.

## 1 INTRODUCTION

Representing the meaning of natural language in a mathematically grounded way is a scientific challenge that has received increasing attention with the explosion of digital content and text data in the last decade. Relying on the richness of contents, several sentence embeddings have been proposed (Peters et al. (2018); Radford et al. (2018); Devlin et al. (2018)) with demonstrated efficiency for the considered tasks. These embeddings, learnt on massive datasets, are commonly used to perform downstream tasks such as classification or data generation.

In a classification context, *e.g.* when the goal is to distinguish between positive and negative sentiments in user reviews, most classifiers are based on Empirical Risk Minimization (ERM) strategies and variants of it. However, nothing guarantees that such classifiers perform satisfactorily on the tails of the explanatory variables, *i.e.* in regions attached to rare events or even outside the domain of observed data. The present paper builds upon the methodological framework proposed by (Jalalzai et al. (2018)) performing classification in extreme regions with guarantees concerning the excess risk of the ERM classifier. Their approach relies on multivariate extreme value theory (EVT) and is valid under regularity assumptions concerning the multivariate distributional tail which appear to be an homogeneity property above large thresholds: the tail behaviour at infinity is similar to the behaviour of polynomial function near infinity (see equation 1 below). To wit, it is necessary for the multivariate survival function of the considered random vector to be approximately homogeneous of degree $-1$, up to appropriate marginal standardization. Although there is experimental evidence that their framework improves the performance of baseline classifiers in extreme regions on low dimensional data, there is no reason to assume that the previously mentioned text embeddings satisfy the required regularity assumptions. The aim of the present work is to apply Jalalzai et al. (2018)'s methodology to text data represented by state of the art embeddings. To achieve this, we train the algorithm to perform a transformation that allows it to map any text embedding $X$ onto a random vector $Z$ which satisfies the aforementioned regularity assumptions. The transformation is learnt by an adversarial strategy (Goodfellow et al. (2016)). As a by-product, we obtain a novel data augmentation mechanism which takes advantage of the scale invariance properties of $Z$ to generate synthetic sentences that keep invariant the attribute of the original sentence.

The work detailed in this paper can be connected to other articles (Socher et al. (2013); Guu et al. (2018); Siffer et al. (2017)), though our framework is different. Our contribution is twofold. We

introduce (*i*) a novel classification algorithm (*Orthrus*), taking as input an embedded sentence to map it onto a heavy-tailed vector in the latent space satisfying Equation 1. (*ii*) a semi-supervised data augmentation method (*Hydra*), which leverages the homothetic invariance property of the embedding learnt by *Orthrus* to produce new sentences with prescribed label.

Label preserving data augmentation, which increases the number of training samples in machine learning systems, is an effective solution to data insufficiency and is an efficient pre-processing step in computer vision (Wang & Perez (2017)). Due to the discrete nature of text, data augmentation is a much harder problem in natural language processing (NLP) than in computer vision. Current work in natural language (NL) dataset augmentation with label preservation mainly relies either on back-translation (Shleifer (2019)), on synonym replacements (Kobayashi (2018)), on slot filling (Hou et al. (2018)) or on the use of handcrafted heuristics (Wei & Zou (2019)) (*e.g* swap, deletion, synonym replacements). Unlike other state of the art solutions, *Hydra* can generate multiple sentences with a considered sentiment using a unique input sentence. *Hydra* does not require any additional resources (e.g. synonym dictionary), nor does it rely on hand-crafted heuristics, or use style transfer methods (Hu et al. (2017); Colombo et al. (2019)).

In this paper, we work with output vectors issued by BERT sentence embedding as inputs (Devlin et al. (2018)). BERT is currently one of the state-of-the art algorithms for sentence embedding. We demonstrate on both synthetic and real datasets that the embedding learnt by *Orthrus* on top of BERT indeed follows a heavy-tail distribution, and that *Orthrus* is able to outperform a state-of-the art classifier built on BERT. On the dataset augmentation task, quantitative and qualitative experiments demonstrate the ability of *Hydra* to generate new sentences while preserving labels.

The rest of this paper is organized as follows. Section 2 introduces the necessary background in multivariate extremes and adversarial learning. The methodology we propose is detailed at length in section 3. Illustrative numerical experiments on both synthetic and real data are gathered in section 4. Additional experimental results are available in the supplementary material.

## 2 BACKGROUND

### 2.1 HEAVY TAILS AND REGULAR VARIATION

By definition, heavy-tail phenomena are those which are ruled by very large values, occurring with a far from negligible probability and with significant impact on the system under study. When the phenomenon of interest is described by the distribution of a univariate random variable, the theory of regularly varying functions provides the appropriate mathematical framework for the study of heavy-tailed distributions. For the sake of clarity and in order to introduce notations we recall some related theoretical background. One may refer to (Resnick (2013)) for an excellent account of the theory of regularly varying functions and its application to the study of heavy-tailed distributions. A random variable $X$ with cumulative distribution function (*c.d.f.*) $F$ is heavy-tailed of index $\alpha > 0$ if and only if for any fixed $x > 0$:

$$n\mathbb{P}\left\{X/F^{-1}(1-1/n) > x\right\} \xrightarrow[n \to \infty]{} x^{-\alpha}$$

where $F^{-1}(u) = \inf\{t : F(t) \geq u\}$ denotes $F$'s generalized inverse. Based on this characterization, the heavy-tail model is classically extended to the multivariate setup as follows. Consider now a $d$-dimensional random vector $X = (X^{(1)}, \ldots, X^{(d)})$ taking its values in $\mathbb{R}^d_+$. Then $X$ is said to be heavy-tailed with tail index $\alpha > 0$ if there exists a positive Radon measure $\mu$ on the punctured set $[0, \infty]^d \backslash \{0\}$ and a function $b(t) \to \infty$ such that:

$$t\mathbb{P}\left\{X/b(t) \in A\right\} \xrightarrow[t \to \infty]{} \mu(A) \tag{1}$$

For any Borelian set $A \subset [0, \infty]^d$ which is bounded away from 0 and such that the measure $\mu$ of the boundary $\partial A$ is zero. In such a case, $\mu$ fulfills the homogeneity property $\mu(tA) = t^{-\alpha}\mu(A)$ for all $t > 0$ and any set $A$ in $[0, \infty]^d \backslash \{0\}$. Using the homogeneity property, one may show that $\mu$ can be decomposed into a radial component and an angular component $\Phi$, which are independent from each other. Indeed, for all $x = (x_1, \ldots, x_d) \in \mathbb{R}^d$, set

$$\begin{cases} R(x) = \|x\| \\ \Theta(x) = \left(\dfrac{x_1}{R(x)}, \ldots, \dfrac{x_d}{R(x)}\right) \in S, \end{cases} \tag{2}$$

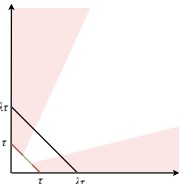

Figure 1: Colored cones correspond to a given label from the classifier on the simplex.

where $S$ is the positive orthant of the unit sphere in $\mathbb{R}^d$ for the chosen norm $\|.\|$. The choice of the norm is unimportant as all norms are equivalent in $\mathbb{R}^d$. Then, for every $B \subset S$, we have:

$$\mu[\{x : R(x)/b(t) > t, \Theta(x) \in B\}] = t^{-\alpha}\Phi(B),$$

where $\Phi$ is referred to as the spectral or angular measure.

## 2.2 CLASSIFICATION IN EXTREME REGION

In the standard setup of binary classification, $(X, Y) \in \mathbb{R}^d_+ \times \{-1, 1\}$ is a random pair defined on a certain probability space with unknown joint probability distribution $P$. Now, equipped with the concepts introduced in the previous section we shall assume regular variation for both classes with the same tail index. Stated otherwise (as formulated in (Jalalzai et al. (2018))) we shall make the assumption that:

$$t\mathbb{P}\left\{X \in tA \mid Y = \sigma 1\right\} \to \mu_\sigma(A)$$

where $\sigma \in \{-1, +1\}$, $A \subset \mathbb{R}^d_+, 0 \notin \partial A$. If one considers $(X_\infty, Y_\infty)$ a random pair defined by $\mathbb{P}\{X_\infty \in A, Y_\infty = +1\} = \lim_{t \to \infty}, \mathbb{P}\{X \in tA, Y = +1 \mid \|X\| > t\}$, $\mathbb{P}\{Y_\infty = +1\} = \lim_{t \to \infty} \mathbb{P}\{Y = +1 \mid \|X\| > t\}$, then one can prove (see Theorem 1 in Jalalzai et al. (2018)) that the Bayes regression function $\eta_\infty(x) = \mathbb{P}\{Y_\infty = 1 | X_\infty = x\}$ for the limiting pair is constant along rays *i.e.* is a function of $\Theta(x) \in S$ only. The same is true of the Bayes classifier for the asymptotic pair. As a consequence, the optimal classifiers on extreme regions are based on indicator functions of truncated cones on the kind $\{\|x\| > r, \Theta(x)B\}$, where $B \subset S$. As a consequence, the optimal classifiers on extreme regions are based on indicator functions of truncated cones on the kind $\{\|x\| > r, \Theta(x) \in B\}$, where $B \subset S$ (see Figure 1). This observation justifies an ERM strategy using the extreme points of the dataset to train an angular classifier.

Let $\{(X_i, Y_i)\}_{i=1}^n$ be n *i.i.d* copies of $(X, Y)$. By sorting the training observations by decreasing order of magnitude, we introduce $X_{(i)}$ (with corresponding sorted label $Y_{(i)}$) the $i$-th order statistics, *i.e.* $\|X_{(1)}\| \geq \ldots \geq \|X_{(n)}\|$. Let $\mathcal{G}_S$ be a class of classifiers defined on the sphere $S$ with finite VC dimension $V_{\mathcal{G}_S} < \infty$. Thus for any $g \in \mathcal{G}_S$ and any $\theta \in S$, $g(\theta) \in \{-1, 1\}$. By extension, for any $x \in \mathbb{R}^d_+$ we define $g(x) = g(\Theta(x)) \in \{-1, 1\}$.

Let $t_\tau$ be the quantile at level $(1 - \tau)$ of the r.v $\|X\|$ where $\tau > 0$ corresponds to a small fraction of extreme observations: $\mathbb{P}\{\|X\| > t_\tau\} = \tau$. Set $k = \lfloor n\tau \rfloor$ and consider the empirical risk dedicated to the extreme samples:

$$\widehat{L}_k(g) = \frac{1}{k}\sum_{i=1}^k \mathbf{1}\{Y_{(i)} \neq g(\Theta(X_{(i)}))\},$$

which corresponds to the empirical version of $L_{t_\tau}$, the loss at level $t_\tau$. Now consider solutions to the minimization problem:

$$\min_{g \in \mathcal{G}_S} \widehat{L}_k(g) \tag{3}$$

Thence the authors of Jalalzai et al. (2018) provide guarantees concerning the classification risk in out-of-sample regions.

**Theorem 1 (Jalalzai et al. (2018))** *Let $\widehat{g}_k$ be any solution of equation 3. Recall $k = \lfloor n\tau \rfloor$. Then, for $\delta \in (0,1)$, $\forall n \geq 1$, we have with probability larger than $1 - \delta$:*

$$L_{t_\tau}(\widehat{g}_k) - L_{t_\tau}^* \leq \frac{1}{\sqrt{k}} \left( \sqrt{2(1-\tau)\log(2/\delta)} + C\sqrt{V_{\mathcal{G}_S}\log(1/\delta)} \right)$$

$$+ \frac{1}{k} \left( 5 + 2\log(1/\delta) + \sqrt{\log(1/\delta)}(C\sqrt{V_{\mathcal{G}_S}} + \sqrt{2}) \right) + \left\{ \inf_{g \in \mathcal{G}_S} L_{t_\tau}(g) - L_{t_\tau}^* \right\},$$

*where $C$ is a constant independent from $n$, $\tau$ and $\delta$.*

### 2.3 ADVERSARIAL LEARNING

Adversarial networks, introduced in (Goodfellow et al. (2014)), form in a system where two neural networks are competing. A first model called the generator generates samples as close as possible to the input dataset. A second model called the discriminator aims at distinguishing samples produced by the generator from the input dataset. The goal of the generator is to maximize the probability of the discriminator making a mistake. Hence, if $P_{\text{input}}$ is the distribution of the input dataset then the adversarial network intends to minimize the distance (as measured by the Jensen-Shannon divergence) between the distribution of the generated data $P_G$ and $P_{\text{input}}$.

Auto-encoders and derivations (Goodfellow et al. (2016); Laforgue et al. (2018); Fard et al. (2018)) shape a subclass of neural networks whose purpose is to learn a suitable representation by learning encoding and decoding functions which capture the core properties of the input data. An adversarial auto-encoders (see Makhzani et al. (2015)) is a specific kind of auto-encoder where the code plays the role of the generator of an adversarial network. Thus the latent code is forced to follow a given distribution while containing information relevant to reconstructing the input.

## 3 DILATION INVARIANT REPRESENTATION

### 3.1 LEARNING A DILATION INVARIANT REPRESENTATION

We now introduce *Orthrus*, a novel algorithm for classification of text data represented by high dimensional vectors as issued by pre-trained embeddings such as BERT which is assumed to not satisfy the regular variation assumption. The idea behind *Orthrus* is to modify the output $X$ of BERT in order to increase the information carried by the resulting representation $Z = \varphi(X)$ regarding the label $Y$, including in the upper tail regions of $Z$ corresponding to low probability sentences, in order to improve the performance of a downstream classifier. This is achieved by training an encoding function $\varphi$ in such a way that *(i)* the marginal distribution $q(z)$ of the code $Z$ be close to a user-specified heavy tailed distribution $p$ satisfying the regularity property (1) ; and *(ii)* the classification loss of a multilayer perceptron trained on the code $Z$ be small. A major difference distinguishing *Orthrus* from existing auto-encoding schemes is that the target distribution on the latent space is not chosen as a Gaussian distribution but as a heavy-tailed one.

As the Bayes classifier in the extreme region has a potentially different structure from the Bayes classifier on the bulk (recall from Section 2 that the regression function at infinity depends on the angle $\Theta(x)$ only), *Orthrus* trains two different classifiers, $C^{\text{ext}}$ on the extreme region of the latent space on the one hand, and $C^{\text{bulk}}$ on its complementary set on the other hand. Given a high threshold $t$, the extreme region of the latent space is defined as the set $\{z : \|z\| > t\}$. In practice, the threshold $t$ is chosen as an empirical quantile of order $(1 - \kappa)$ (for some small, fixed $\kappa$) of the norm of encoded data $\|Z_i\| = \|\varphi(X_i)\|$. In the end, the classifier issued by *Orthrus* is of the kind $C(z) = C^{\text{ext}}(z)\mathbf{1}\{\|z\| > t\} + C^{\text{bulk}}(z)\mathbf{1}\{\|z\| \leq t\}$ . Our goal is to minimize the weighted risk

$$R(\varphi, C^{\text{ext}}, C^{\text{bulk}}) = \lambda_1 \mathbb{P}\left\{ Y \neq C^{\text{ext}}(Z), \|Z\| \leq t \right\} + \lambda_2 \mathbb{P}\left\{ Y \neq C^{\text{bulk}}(Z), \|Z\| > t \right\}$$
$$+ \lambda_3 \mathfrak{D}(q(z), p(z)), \tag{4}$$

where $Z = \varphi(X)$, $d$ is the Jensen-Shannon distance between the heavy tailed target distribution $p$ and the code distribution $q$, and $\lambda_1, \lambda_2, \lambda_3$ are positive weights. Following common practice in the adversarial literature, the Jensen-Shannon distance is approached (up to a constant term) by the

empirical proxy $\widehat{L}(q,p) = \sup_{D \in \Gamma} \widehat{L}(q,p,D))$, with

$$\widehat{L}(q,p,D) = \frac{1}{m} \sum_{i=1}^{m} \log D(Z_i) + \log(1 - D(\tilde{Z}_i))$$

where $\Gamma$ is a wide class of discriminator functions, and where independent samples $Z_i, \tilde{Z}_i$ are respectively sampled from the target distribution and the code distribution $q$. In the end, *Orthrus* solves the following min-max problem $\inf_{C^{\text{ext}}, C^{\text{bulk}}, \varphi} \sup_D \widehat{R}(\varphi, C^{\text{ext}}, C^{\text{bulk}}, D)$ where

$$\widehat{R}(\varphi, C^{\text{ext}}, C^{\text{bulk}}) = \frac{\lambda_1}{k} \sum_{i=1}^{k} 1\{Y_{(i)} \neq C^{\text{ext}}(Z_{(i)})\} + \frac{\lambda_2}{n-k} \sum_{i=1}^{n-k} 1\{Y_{(i)} \neq C^{\text{bulk}}(Z_{(i)})\} + \cdots$$

$$\cdots \lambda_3 \, \widehat{L}(q,p)),$$

where $\{Z_{(i)} = \varphi(X_{(i)}), i = 1, \ldots, n\}$ are the encoded observations with associated labels $Y_{(i)}$ sorted by decreasing magnitude of $\|Z\|$ (*i.e.* $\|Z_{(1)}\| \geq \cdots \geq \|Z_{(n)}\|$), and $k = \lfloor \kappa n \rfloor$ is the number of extreme samples among the $n$ encoded observations.

This minimization problem results in Algorithm 1 detailed below, where $m$ is the batch size, $\rho$ is the proportion of the points given to train the extreme classifier and the cost function $\ell$ is the negative log-likelihood.

## 3.2 A DILATION INVARIANT REPRESENTATION FOR DATASET AUGMENTATION

We now introduce *Hydra*, a data augmentation algorithm, which relies on the dilation invariance of labels on the extreme embeddings learnt by *Orthrus*. Let $U = (u_1, \ldots, u_T)$ be a sequence of inputs of length $T$. The dataset augmentation problem consists in generating multiple sentences $\{U_i'\}_{i \in \mathbb{N}}$ such that each sequence $U_i'$ is label coherent with $U$. *Hydra* follows the seq2seq approach (Sutskever et al. (2014)) and learns two decoders with attention (Bahdanau et al. (2014)), one for each region of the representation previously learnt by *Orthrus*. On the whole, a decoder takes an input $Z$ (the latent code) and generates an output sequence $U_i' = (u_{i_1}', \ldots, u_{i_{T'}}')$, where each word $u_{i_k}'$ is in the vocabulary. To generate an output word $u_k'$, the decoder iteratively takes as input the previously generated word $u_{k-1}'$ ($u_0'$ being a start symbol), updates its internal state, and returns the next word with highest probability. This process is repeated until the decoder generates either a stop symbol or the length of the generated sequence reaches the maximum sequence length.

*Hydra* is a semi-supervised algorithm in that it requires a first labeled dataset $\mathcal{D}_n$ to learn embeddings through *Orthrus* and a second dataset $\mathcal{D}_{g_n}$ (not necessarily labeled) to train a decoder $G^{\text{ext}}$ (*resp* $G^{\text{bulk}}$) for the extremes (*resp* non extremes) on $\mathcal{D}_{g_n}$. The learning is carried out by optimising the classical negative log likelihood of individual tokens $\ell_{gen}$. A detailed description of *Hydra* is provided in Algorithm 2 below.

After *Hydra* has been trained, the data augmentation proceeds as follows: For each input $X$ with extreme embedding, multiple sentences with constant label are obtained by applying a $\lambda$ dilation (see Figure 2b). In other words, if $G^{\text{ext}}$ is a decoder learnt by *Hydra* and applicable to the extreme region of the representation, synthetic sentences of the form $G^{\text{ext}}(\lambda z)$ can be obtained, with the same attribute as $z$.

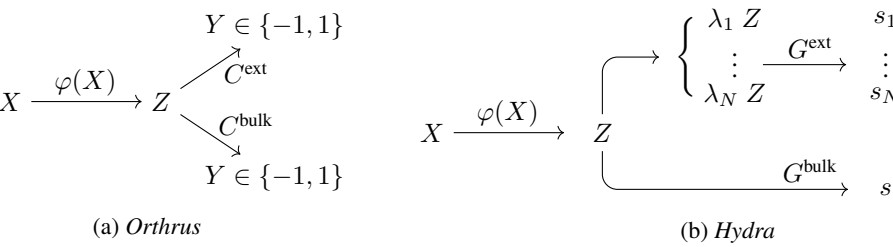

(a) *Orthrus*  (b) *Hydra*

Figure 2: Pipeline of the introduced algorithms.

---

**Algorithm 1** Orthrus

---

**INPUT:** Coef. $\lambda_1, \lambda_2, \lambda_3 > 0$, Training dataset $\mathcal{D}_n = \{(U_1, Y_1), \ldots, (U_n, Y_n)\}$, proportion $\kappa \leq 1$ of extreme observations, batch size $m \leq n$.

**Initialization:** parameters of the encoder $\varphi_\tau$, classifiers, $C_\theta^{ext}, C_{\theta'}^{bulk}$, decoders and discriminator $D_\gamma$

   **while** $(\tau, \theta, \theta')$ not converged **do**
   Sample $\{(U_1, Y_1) \ldots, (U_m, Y_m)\}$ from $\mathcal{D}_n$.
   Sample $\{Z_1, \ldots, Z_m\}$ from the prior $P_Z$.
   Sample $\tilde{Z}_i$ from $\varphi_\tau(Z|X_i)$ for $i \in \{1, \ldots, m\}$.

   Update $D_\gamma$ by ascending:

$$\frac{\lambda_3}{m} \sum_{i=1}^{m} \log D_\gamma(Z_i) + \log(1 - D_\gamma(\tilde{Z}_i)).$$

   Sort $\{\tilde{Z}_i\}_{i \in \{1, \ldots, m\}}$ by decreasing order of magnitude $||\tilde{Z}_{(1)}|| \geq \ldots \geq ||\tilde{Z}_{(m)}||$.
   Update $C_\theta^{ext}$ by descending:

$$\mathcal{L}^{ext} \stackrel{\text{def}}{=} \frac{\lambda_1}{\lfloor \kappa m \rfloor} \sum_{i=1}^{\lfloor \kappa m \rfloor} \ell\big(Y_{(i)}, C_\theta^{ext}(\tilde{Z}_{(i)})\big).$$

   Update $C_{\theta'}^{bulk}$ by descending:

$$\mathcal{L}^{bulk} \stackrel{\text{def}}{=} \frac{\lambda_2}{m - \lfloor \kappa m \rfloor} \sum_{i=\lfloor \kappa m \rfloor + 1}^{m} \ell\big(Y_{(i)}, C_{\theta'}^{bulk}(\tilde{Z}_{(i)})\big).$$

   Update $\varphi_\tau$ by descending:

$$\frac{1}{m} \sum_{i=1}^{m} -\lambda_3 \cdot \log D_\gamma(\tilde{Z}_i) + \mathcal{L}^{ext} + \mathcal{L}^{bulk}.$$

   **end while**
   Compute $\{\tilde{Z}_i\}_{i \in \{1, \ldots, n\}} = \varphi(X_i)_{i \in \{1, \ldots, n\}}$
   Sort $\{\tilde{Z}_i\}_{i \in \{1, \ldots, n\}}$ by decreasing order of magnitude $||\tilde{Z}_{(1)}|| \geq \ldots ||\tilde{Z}_{(\lfloor \kappa n \rfloor)}|| \geq \ldots \geq ||\tilde{Z}_{(n)}||$.

**OUTPUT:** encoder $\varphi$, classifiers $C_\theta^{ext}$ applicable on the region $\{x : ||\varphi(x)|| \geq ||\tilde{Z}_{(\lfloor \kappa n \rfloor)}||\}$ and $C_\theta^{bulk}$ applicable on the region $\{x : ||\varphi(x))|| < ||\tilde{Z}_{(\lfloor \kappa n \rfloor)}||\}$.

---

**Algorithm 2** Hydra

---

**INPUT:** Coef. $\lambda_1, \lambda_2, \lambda_3 > 0$, Training datasets $\mathcal{D}_n = \{(U_1, Y_1), \ldots, (U_n, Y_n)\}$, $\mathcal{D}_{g_n} = \{U_{g_1}, \ldots, U_{g_n}\}$ batch size $m \leq n$,

**Initialization:** parameters of the encoder $\varphi$, classifiers, $C_\theta^{ext}, C_{\theta'}^{bulk}$, decoders, $G_\psi^{ext}, G_{\psi'}^{bulk}$ and discriminator $D_\gamma$

**Optimization:**
   $\varphi, C^{ext}, C^{bulk} = Orthrus(\lambda_3, \mathcal{D}_n,)$

   **while** $(\psi, \psi')$ not converged **do**
   Sample $\{(X_{g_1}) \ldots, (X_{g_m})\}$ from the training set $\mathcal{D}_{g_n}$.
   Sample $\tilde{Z}_i$ from $\varphi(Z|X_{g_i})$ for $i \in \{1, \ldots, m\}$.

   Sort $\{\tilde{Z}_i\}_{i \in \{1, \ldots, m\}}$ by decreasing order of magnitude

$$||\tilde{Z}_{(1)}|| \geq \ldots \geq ||\tilde{Z}_{(m)}||.$$

   Update $G_\psi^{ext}$ by descending:

$$\mathcal{L}_g^{ext} \stackrel{\text{def}}{=} \frac{\lambda_1}{\lfloor \rho m \rfloor} \sum_{i=1}^{\lfloor \rho m \rfloor} \ell_{gen.}\big(Y_{(i)}, G_\psi^{ext}(\tilde{Z}_{(i)})\big).$$

   Update $G_{\psi'}^{bulk}$ by descending:

$$\mathcal{L}^{bulk} \stackrel{\text{def}}{=} \frac{\lambda_2}{m - \lfloor \rho m \rfloor} \sum_{i=\lfloor \rho m \rfloor + 1}^{m} \ell_{gen.}\big(Y_{(i)}, G_{\psi'}^{bulk}(\tilde{Z}_{(i)})\big).$$

   **end while**
   Compute $\{\tilde{Z}_i\}_{i \in \{1, \ldots, n\}} = \varphi(X_i)_{i \in \{1, \ldots, n\}}$

   Sort $\{\tilde{Z}_i\}_{i \in \{1, \ldots, n\}}$ by decreasing order of magnitude $||\tilde{Z}_{(1)}|| \geq \ldots ||\tilde{Z}_{(k)}|| \geq \ldots \geq ||\tilde{Z}_{(n)}||$.

**OUTPUT:** encoder $\varphi$, decoder $G^{ext}$ applicable on the region $\{x : ||\varphi(x)|| \geq ||\tilde{Z}_{(\lfloor \kappa n \rfloor)}||\}$ and $G^{bulk}$ applicable on the region $\{x : ||\varphi(x)|| < ||\tilde{Z}_{(\lfloor \kappa n \rfloor)}||\}$.

---

Algorithm: *Orthrus* and *Hydra*. For both algorithm $X$ is the embedding provided by BERT to an input sentence $U$.

## 4 NUMERICAL EXPERIMENTS

**Heavy-tail distribution:** In the remaining of this paper, the regularly varying target distribution is a logistic distribution with parameter $\delta = 0.9$ (see Appendix B.2). The logistic distribution is widely used in the context of extreme values analysis. This distribution is simulated according to Stephenson (2003) which is defined in $\mathbb{R}^d$ with parameter $\delta \in (0, 1]$ by its parametric *c.d.f.* $F(x) = \exp\left\{-\left(\sum_{j=1}^{d} x^{(j)\frac{1}{\delta}}\right)^{\delta}\right\}$. Figure 4 in Appendix illustrates this distribution with various values of $\delta$.

**Experimental Settings:** Classifiers $C^{\text{bulk}}, C^{\text{ext}}$ are Multi Layer Perceptrons (MLP). For each dataset, one-fourth of data is retained as a test set while the remaining data points are used as the train set. We denote by $\mathcal{T}$ the extreme test set region as selected with Algorithm 1. The considered norm is the $\ell_\infty$ norm. For the generation task, the dilation coefficient varies between 1 and 1.5 ($\lambda \in [1, 1.5]$).

### 4.1 APPLICATION TO SENTIMENT ANALYSIS

We assess *Othrus* ability to perform sentiment classification tasks using two balanced binary labeled datasets: Yelp, Amazon introduced by (Kotzias et al. (2015)). Further results on the datasets are reported in the Appendix. Each dataset is balanced and is composed of 1000 sentences and each sentence expresses a strong positive or negative sentiment.

**Baseline:** *Orthrus* is compared with a state of the art classifier: a MLP trained on the initial BERT embedding ($NN$) taking as input the same training data.

**Classification performance:** Table 1 illustrates the performance of considered models in terms of Hamming loss on the test set. Performance of *Orthrus* is better than $NN$ on both dataset on the whole test set. The difference in terms of performance is even greater when considering the extreme samples.

| Model | Hloss All | Hloss Extreme | | Model | Hloss All | Hloss Extreme |
|-------|-----------|---------------|---|-------|-----------|---------------|
| NN | 0.30 | 0.30 | | NN | 0.33 | 0.32 |
| *Orthrus* | 0.26 | 0.24 | | *Orthrus* | 0.29 | 0.22 |
| (a) | | | | (b) | | |

Table 1: Hamming loss (Hloss) on the Yelp (Table 1a) and Amazon (Table 1b) datasets. The first column transcribes the loss on the whole test set while the second column transcribes the loss on the extreme samples.

### 4.2 APPLICATION TO DATASET AUGMENTATION FOR LABELLED SENTENCES

**Comparison to related work:** We compare *Hydra* with two state of the art methods for dataset augmentation: EDA (Wei & Zou (2019)) and round-trip translation (Shleifer (2019)). Compared to EDA which uses heuristics and a synonym dictionary, *Hydra* generates grammatically correct sentences and conserve the labels. Compared to round-trip translation *Hydra* can generate multiple sentences for a single input and conserves sentence labels. We also compare *Hydra* to a Vanilla Sequence to Sequence to demonstrate the validity of our approach. In this section, we validate *Orthrus* using the Yelp dataset. To train *Orthrus* the dataset $\mathcal{D}_n$ is the Yelp dataset introduced in (Kotzias et al. (2015)). $\mathcal{D}_{g_n}$ is a subset of the full Yelp Corpus composed of $5 \cdot 10^5$ sentences.

**Evaluation:** Automatic evaluation of generative models for text is still an open research problem. We evaluate our models through three criteria (**C1, C2, C3**). **C1** measures Cohesion (Crossley & McNamara (2010)) (Are the generated sentences grammatically and semantically consistent?). **C2** (named Sent. in Table 1a.) evaluates label conservation (Does the expressed sentiment in the generated sentence match the sentiment of the input sentence?). **C3** measures the diversity of the sentences (Is the dataset augmented with diverse sentences?)

We carry out two types of evaluations: a qualitative one based on a user study – that provides evaluation through C1 and C2 – and a quantitative one – that provides evaluation through C3 measures. We measure diversity (**C3**) with two automatic methods: (1) *Distinct n*, the diversity measure introduced by (Li et al. (2015)) which relies on a n-grams count; (2) a F1 score that reports the improvement induced by the data augmentation while training a fastText (Joulin et al. (2016)) classifier. This improvement reflects the information added by newly generated sentences.

| Model | Sent | Cohesion |
|-------|------|----------|
| Raw | 0.90 | 0.85 |
| Back | 0.78 | 0.84 |
| Eda | 0.87 | 0.68 |
| Seq2seq | 0.60 | 0.87 |
| *Hydra* | 0.88 | 0.90 |

(a)

| Model | F1 | Distinct 1 | Distinct 2 |
|-------|------|-----------|-----------|
| Raw | 0.87 | 0.385 | 0.817 |
| Back | 0.87 | 0.405 | 0.816 |
| Eda | 0.95 | 0.387 | 0.813 |
| Seq2seq | 0.807 | 0.375 | 0.813 |
| *Hydra* | 0.95 | 0.457 | 0.844 |

(b)

Table 2: Evaluation of *Hydra* with user evaluation (Table 1a) and with automatic metrics (Table 1b).

| input | penne vodka excellent! |
|-------|------------------------|
| $\lambda = 1$ | penne vodka splendid! |
| $\lambda = 1.2$ | the penne vodka is excellent! |
| $\lambda = 1.3$ | the penne vodka is excellent! |
| $\lambda = 1.5$ | the penne vodka is perfect! |
| input | awful service. |
| $\lambda = 1$ | bad maintenance. |
| $\lambda = 1.1$ | a terrible service. |
| $\lambda = 1.3$ | it's a terrible service. |
| $\lambda = 1.5$ | horrible service. |

| input | i love their fries and their beans. |
|-------|-------------------------------------|
| $\lambda = 1$ | i love their fries and their bananas. |
| $\lambda = 1.1$ | i like their pies and their bananas. |
| $\lambda = 1.3$ | i enjoy her fries and her beans. |
| $\lambda = 1.5$ | i enjoy enjoy enjoy and beans. |
| input | awesome selection of beer. |
| $\lambda = 1$ | great selection of beer. |
| $\lambda = 1.1$ | a superb selection of beer. |
| $\lambda = 1.3$ | an amazing selection of beer. |
| $\lambda = 1.5$ | great choice of beer. |

Table 3: Sentences generated by *Hydra* for extreme embeddings implying label (sentiment) invariance for generated sentence. $\lambda$ is the dilation factor.

**Qualitative Evaluation**: We compare the different methods on the extreme set of sentences according to *Hydra*. As previously, the extreme test set $\mathcal{T}$ consists of 92 sentences. For each sentence, we randomly sample $\lambda \in [1, 1.5]$ and generate a new sentence with *Hydra*. To compare our method, we use Eda, round-trip translation and a Vanilla Seq2seq to generate new sentences. Results of the qualitative evaluation are reported in Table 2a. As expected *Hydra* exhibits comparable performance with all other models except Eda in terms of cohesion. Such a result is expected since Eda's use of random swaps or permutations may result in ungrammatical sentences. In terms of expressed sentiments, *Hydra* outperforms Seq2seq and round-trip translation, demonstrating the relevance of the proposed approach.

**Quantitative Evaluation**: We report in (Table 2b) the results of evaluation for **C3**. We observe that *Hydra* outperforms all other methods in term of distinct 1 and distinct 2. Table 1b shows that improvement in F1 score induced by dataset augmentation by *Hydra* beats all other methods and is only equalled by EDA.

**Example of generated sentences**: We report in Table 3 sentences generated by *Hydra*. While we make no claim that these generated sentences are better than sentences generated by existing methods, we believe that these samples are at least competitive with the generative models in the literature and highlight the potential of the introduced framework.

## 5 DISCUSSION & PERSPECTIVES

The approach promoted in this paper relies on learning a regularly varying representation by minimizing an empirical proxy of an objective function. The latter writes as a weighted sum of a classification risk and a regularization term penalizing the distance between the representation distribution and a heavy-tailed target. Our experiments show that the obtained representation allows to diminish the classification error compared to a MLP with comparable complexity.The obtained representation used for a text data augmentation task, is competitive with existing data augmentation methods. The attribute invariance under code dilation is the key to generate meaningful sentences with prescribed attribute. Future work will investigate the possibility to train an auto-encoder satisfying this dilation invariance property without relying on a pretrained classifier such as *Orthrus*.

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

# APPENDIX

## A    REMARKS

**Remark 1** (**Selection of** $k$) To the best of our knowledge, selection of $k$ in Algorithm 1 and Algorithm 2 is still a vivid problem in EVT which is not solved yet. As $k$ gets large the number of extreme points increases including samples which are not large enough and deviates from the asymptotic distribution of extremes. Smaller values of $k$ increase the variance of the classifier/generator. This bias-variance trade-off is beyond the scope of this paper.

**Remark 2** In Figure 3a selecting the extreme samples on the input space is not a straightforward step since the tails of the input clusters are not heavy-tail. It was decided to standardise the input data $(X_i)_{i \in \{1, \cdots, n\}}$ by applying the rank-transformation:

$$\widehat{T}(x) = \left( 1 / \left( 1 - \widehat{F}_j(x) \right) \right)_{j=1, \cdots, d}$$

for all $x = (x^1, \cdots x^d) \in \mathbb{R}^d$ where $\widehat{F}_j \stackrel{\text{def}}{=} \frac{1}{n+1} \sum_{i=1}^n 1\{X_i^j \leq x\}$ is the $j^{th}$ empirical marginal distribution. It comes $\forall i \in \{1, \cdots, n\}, V_i = \hat{T}(|X_i|)$. $V_i$ margins follow a standard Pareto distribution. Extreme samples are thus $\{V_i, ||V_i|| \geq n/k\}$ with $k = \sqrt{n}$ by convention.
One may notice that extremes in Figure 3c are more label balanced compared to Figure 3a.

## B    EXPERIMENTS

Further synthetic experiments were conducted to show the improvement of the classifier.

### B.1    SYNTHETIC DATA

**Dataset description**: Synthetic data was generated using Scikit-Learn (Pedregosa et al. (2011)). Focus is made on the classification dataset where clusters of points are normally distributed for each class (see Figure 3a).
**Convergence to a logistic**: In Figure 3b we visualise the latent space of the output of the encoder $\varphi$ (see Figure 2). In *Orthrus*, the adversarial approach to minimize the discrepancy between the distribution of the output of the encoder and a logistic distribution. Notice the similarity between Figure 3b and Figure 4a.
**Extremes selection:** Extreme samples from Figure 3a are assessed directly on the input data while extremes from Figure 3c are obtained from the latent space. Extremes from the latent space (output of $\varphi$ after the training phase) are located on the border of the distribution which comforts the correct behaviour of the training. We also note similarities while comparing in the input space: extreme samples selected in the input space and extreme ones selected in the learnt latent space. In Figure 3a no extreme observation appear on the negative class whereas labels of extremes of Figure 3c are balanced.

### B.2    LOGISTIC DISTRIBUTION

Figure 4 illustrates the logistic distribution with varying parameter $\delta$.

### B.3    ADDITIONAL EXPERIMENT SETTINGS FOR REAL DATA

We use BERT pretrained models and code from the library *pytorch-transformers*. All models were implemented using Pytorch. The output of BERT is a $\mathbb{R}^{768}$ vector, $Z$ is a $\mathbb{R}^{50}$ vector. The dimension of $Z$ (50) was obtained by cross validation. $D$, $C^{\text{bulk}}$ and $C^{\text{ext}}$ are MLP composed of 3 layers and uses a ReLu as activation function. Decoders $D^{bulk}$ and $D^{ext}$ are GRU composed of 6 layers. For training the batch size is set to 64 and we set $\lambda_1 = (1 - \widehat{\mathbb{P}}(||Z|| \geq ||Z_{\lfloor \kappa n \rfloor}||))^{-1}, \lambda_2 = \widehat{\mathbb{P}}(||Z|| \geq ||Z_{\lfloor \kappa n \rfloor}||)^{-1}, \lambda_3 = 10^{-3}$. All neural networks have been optimized with Adam (Kingma & Ba (2014)). For generation the maximum dilation factor $\lambda$ is set to 1.5 since we observe that with greater values a stuttering phenomenon appears.

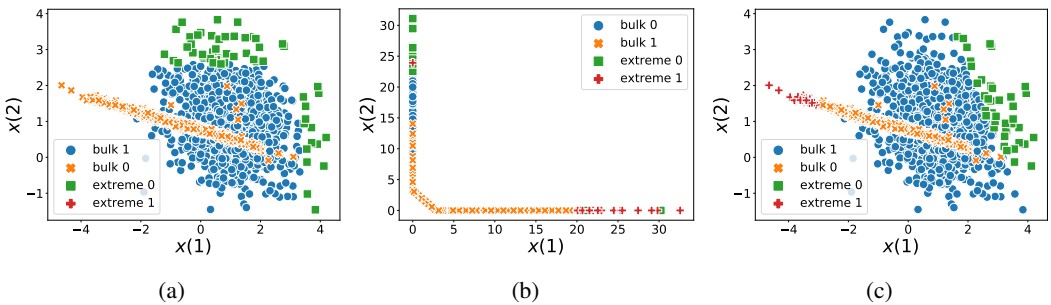

Figure 3: Figure 3a: Bivariate synthetic samples normally distributed and designed for binary classification. Extreme observations are based on a standardisation of the observations in the input space (refer to the Supplementary for details about the standardisation). Figure 3b: Bivariate outputs of the encoder $\varphi$ on the dataset visualised in Figure 3a. Figure 3c Input space where extreme observations are based on $||Z||$ where $Z$ is the output of $\varphi$.

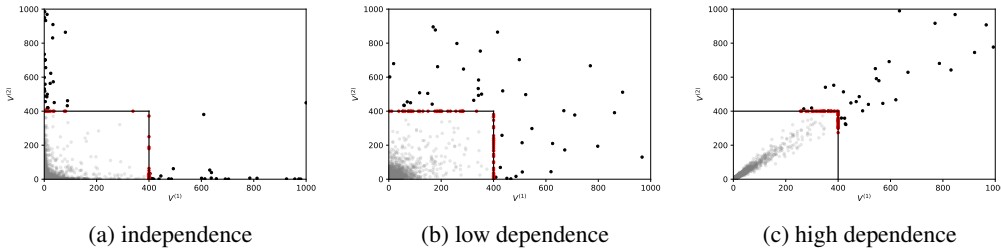

Figure 4: Illutration of the direction $\Theta(V)$ obtained with bivariate samples $V$ generated from a logistic model with different coefficients of dependence from no dependence Figure 4a ($\delta = 0.9$) to high dependence Figure 4c ($\delta = 0.1$) through low dependence Figure 4b ($\delta = 0.5$) . Non extreme samples are in gray, extreme samples are in black and directions $\Theta(V)$ (extreme samples projected on the sup norm sphere) in red. Note that not all extremes are shown since the plot was truncated for a better visualization. However all projections on the sphere are shown.

### B.4 ADDITIONAL EXPERIMENTS ON YELP DATA

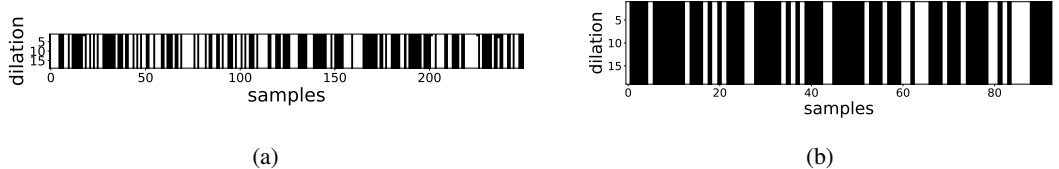

Figure 5: Evolution of the prediction of *Orthrus* on the whole test set (a) and on the extreme test set (b). Colors (black and white) represent predicted labels for each sample ($\lambda \, x_{\text{test}}$) with $\lambda$ being the dilation factor (Y-axis).

**Dilation invariance of $C^{\text{ext}}$ :** Figure 5a shows the evolution of the prediction for each sample of the test set applying the dilation. Predictions of $C^{\text{bulk}}$ change for some samples resulting in bichromatic columns. Figure 5b focuses on the samples from $\mathcal{T}$ and is exclusively composed with monochromatic columns demonstrating that the predicted labels from $C^{\text{ext}}$ do not change, $C^{\text{ext}}$ is homothetic invariant. With no assumption on $C^{\text{ext}}$, this classifier enjoys favorable theoretical properties since it belongs to the class of angular classifiers at the end of the training phase.

| input | ( it wasn't busy either), the building was cold. |
|---|---|
| $\lambda = 1$ | (it was not occupied either), the building was cold. |
| $\lambda = 1.1$ | (i wasn't busy either), the building was frozen. |
| $\lambda = 1.3$ | also, the building was freezing. |
| $\lambda = 1.5$ | plus, the building was colder than ice. |
| input | food quality has been horrible . |
| $\lambda = 1$ | food quality has been terrible. |
| $\lambda = 1.1$ | the quality of the food was horrible. |
| $\lambda = 1.3$ | the quality of the food has been horrible. |
| $\lambda = 1.5$ | the quality of food was terrible. |
| input | overall , i like there food and the service . |
| $\lambda = 1$ | i love food and the service. |
| $\lambda = 1.1$ | on the whole, i like food and service. |
| $\lambda = 1.3$ | in general, i like the food and the service. |
| $\lambda = 1.5$ | in general, i like food and service. |
| input | the desserts were a bit strange . |
| $\lambda = 1$ | the desserts were a little weird. |
| $\lambda = 1.1$ | the desserts were very strange. |
| $\lambda = 1.3$ | the desserts were terrible. |
| $\lambda = 1.5$ | the desserts were terrible. |
| input | we definately enjoyed ourselves . |
| $\lambda = 1$ | we enjoyed ourselves. |
| $\lambda = 1.1$ | we had a lot of fun. |
| $\lambda = 1.3$ | we've really enjoyed each other. |
| $\lambda = 1.5$ | we certainly had fun. |

| input | seriously killer hot chai latte. |
|---|---|
| $\lambda = 1$ | -it's a real killer. |
| $\lambda = 1.2$ | he is a real killer. |
| $\lambda = 1.3$ | he likes to kill. |
| $\lambda = 1.5$ | i loves murders. |
| input | all of the tapas dishes were delicious ! |
| $\lambda = 1$ | all the tapas was delicious. |
| $\lambda = 1.1$ | all tapas dishes were delicious! |
| $\lambda = 1.3$ | all the tapas dishes were delicious! |
| $\lambda = 1.5$ | the tapas were great! |
| input | there was hardly any meat. |
| $\lambda = 1$ | there was almost no meat. |
| $\lambda = 1.1$ | there was practically no meat. |
| $\lambda = 1.3$ | there was almost no meat. |
| $\lambda = 1.5$ | there was no meat. |
| input | waiter was a jerk. |
| $\lambda = 1$ | the waiter was a jerk. |
| $\lambda = 1.1$ | awaiter was a poor guy. |
| $\lambda = 1.3$ | waiter was an idiot. |
| $\lambda = 1.5$ | waiter was such an idiot. |
| input | i 'm not eating here ! |
| $\lambda = 1$ | i don't eat here. |
| $\lambda = 1.1$ | i don't eat here! |
| $\lambda = 1.3$ | i'm not going to eat here! |
| $\lambda = 1.5$ | i will never going to eat here! |

Table 4: Supplementary sentences generated by *Hydra* for extreme embeddings implying label (sentiment) invariance for generated sentence. $\lambda$ is the dilation factor.

### B.5 ADDITIONAL EXPERIMENTS ON AMAZON DATA

Experiments on the Amazon dataset were conducted similarly to to the ones from the Yelp dataset from subsection 4.1, similar conclusions can be drawn from Figure 6.

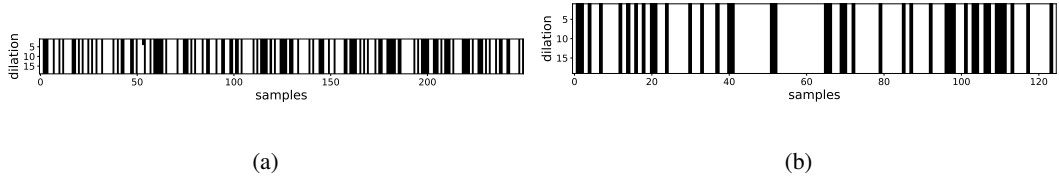

(a)  (b)

Figure 6: Evolution of the prediction of *Orthrus* on the whole test set (a) and on the extreme test set (b). Colors represent predicted labels for each sample ($\lambda\, x_{\text{test}}$) with $\lambda$ being the dilatation factor (Y-axis).

### B.6 QUALITATIVE EVALUATION

For the qualitative evaluation, we compute the Krippendorff's alpha coefficient Krippendorff (2018) to evaluate the inter-annotator agreement. We obtain $\alpha = 0.41$ for the grammar analysis and $\alpha = 0.53$ for the sentiment analysis. If $\alpha = 1$ it indicates perfect reliability; if $\alpha = 0$ it indicates the absence of reliability. Such values of $\alpha$ can be considered as relatively high given our huge number of annotators (126 for the grammar and 134 for the sentiment analysis).

### B.7 ADDITIONAL GENERATED SENTENCES

We present in Table 4 additional extreme sentences.

## C A LINGUISTIC POINT OF VIEW OF EXTREMES

In this section, we provide a comparative linguistic analysis of the extreme samples in case of sentiment classification through the affect scores provided by the lexicon for English words from

Mohammad (2017).

To better understand the content of extreme and non extreme text samples in experiments conducted in section 4, the *maximum* and the *minimum* of the affect score for each sentence are studied (see Figure 7).

For both Yelp and Amazon datasets, one can notice that the violin plot of the *maximum* of the affect score is wider among the extremes than in the bulk. This illustrates that extreme tend to have both a more positive and a more negative affect score than the non extreme observations. Same goes for the *minimum* affect score of extremes whose violin plot tends to be larger among the extremes compared to the violin plot of the *minimum* affect score on the bulk.

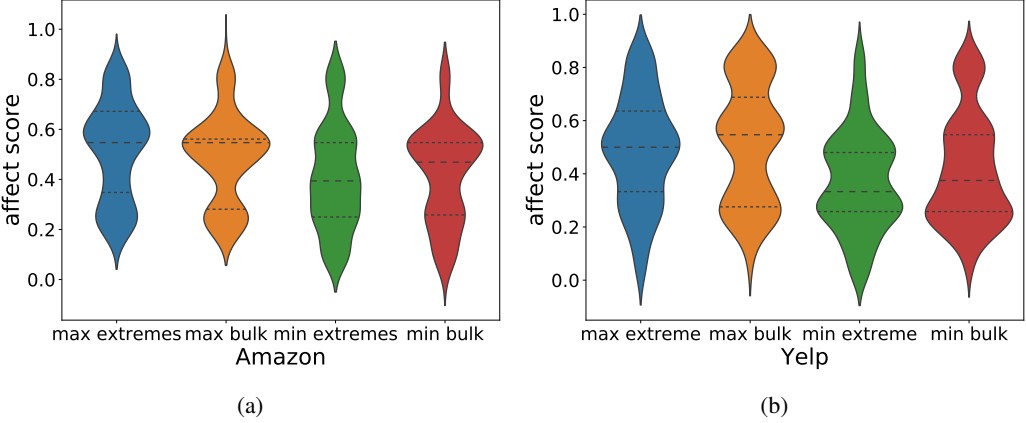

(a)                                                     (b)

Figure 7: Illutration of the affect score on the Amazon dataset (Figure 7a) and on the Yelp dataset (Figure 7b dataset). The blue (respectively orange) violin plot represents the *maximum* affect score of sentences in the extreme (respectively non extreme) samples. The green (respectively red) violin plot represents the *minimum* affect score of sentences in the extreme (respectively non extreme) samples.

