# OpenReview forum: "Regularly varying representation for sentence embedding"
_ICLR.cc/2020/Conference — Reject_

### Official Review · AnonReviewer3 · 2019-10-21
**Official Blind Review #3**

**Rating:** 3

**Review:**

The paper presented two methods for augmenting sentiment classification from the perspective of applying the Extreme Value Theory (EVT), including:

1) A classification algorithm which has an adversarial classifier to enforce the intermediate representations of a neural network to be similar to one EVT distribution, logistic distribution;
2) An encoder-decoder model that is able to generate grammatically coherent sentences with the same sentiment as the given input sentence.

Questions:

--
1) the Fisher–Tippett–Gnedenko theorem states that it is possible that the maximum value of a set of iid samples converges to one of three plausible distributions, and the chosen logistic distribution falls into the Weibull distribution category. I have a couple concerns about this choice:

1.1) In order to show that the EVT indeed helps empirically in the way that an adversarial classifier enforces the inf-norm of vectors follow the Generalised Extreme Value (GEV) distribution, at least three plausible distributions from each form of the GEV distribution needs to be checked. The logistic distribution is interesting, but the marginal improvement gained by enforcing the lengths of the produced vectors to follow the logistic distribution could be a result of hyper-param tuning, which shouldn't be a piece of supporting evidence.

1.2) From the perspective of applying the EVT, recent successful work from the best of my knowledge is on Anomaly Detection [1], where the EVT enables the system to learn from samples in only one class and also adjust the threshold for detecting the abnormal behaviour of samples. It is also theoretically grounded as the error variable of a logistic regression follows a Gumbel distribution which is one form of the GEV distribution, therefore, applying EVT for binary classification case makes sense.

1.3) From the perspective of learning representations with structured priors, there exists an interesting work on decomposing vector representations into lengths and directions and enforcing lengths to follow a uniform distribution and directions a Von Mises–Fisher (vMF) distribution as in [2]. It would be interesting to see if the proposed method is indeed better than the way that structured priors are enforced in [2].

1.4) Linguistically, given the distributional hypothesis, the length of learnt vectors tends to be highly correlated with the frequency information of available concepts and the direction of them matters more. The argument is also presented by the paper. However, in sentiment analysis, the length could contain the information about how strong the sentiment of the input sentence is, so I am not convinced that the proposed method would be applicable in fine-grained sentiment analysis, such as Stanford Sentiment Treebank [3].


--
2) A soft approximation over the inf-norm of a set of iid samples is log-sum-exp function, and it is the cdf of softmax function, which is also theoretically grounded in EVT for classifications. It could be a nicer story than the current one as the choice of the logistic distribution seems to be too intend.


--
3) The construction of the two datasets seems to be very arbitrary given that there exists a large number of sentiment analysis datasets and many with lots of samples, I am not sure that the results on the chosen constructed two datasets are sufficient enough to support the claim.

3.1) The size of the datasets is too small. Given that, the marginal improvement against the NN baseline could be a result of a specific initialisation, which doesn't generalise to other random initialisations.

3.2) The dimension of vector representations is also too small. Normally, commonly used word embeddings are of 300 dimensions, and contextualised ones are of higher than 1200 dimensions. The chosen 50 dimension could prevent the NN baseline model to perform well and IMO, it is helpful for picking a suitable logistic prior than it is in a very high dimensional space.

3.3) There are many straightforward distributions that could be applied as a prior on the lengths of vector representations, e.g. the Rayleigh distribution in 2D and the Chi-squared distribution in higher-dimension. Then again, the distribution gets flatter and becomes similar to a uniform distribution when the dim goes higher, which is a common issue. It goes back to my concern or doubt on the usability of a prior on the norm of high dimensional vectors.


--
4) I am still interested in seeing EVT being applied in various domains, but I'd be in favor of more justifiable approaches.

[1] Siffer, Alban, et al. "Anomaly detection in streams with extreme value theory." Proceedings of the 23rd ACM SIGKDD International Conference on Knowledge Discovery and Data Mining. ACM, 2017.
[2] Guu, Kelvin, et al. "Generating sentences by editing prototypes." Transactions of the Association for Computational Linguistics 6 (2018): 437-450.
[3] Socher, Richard, et al. "Recursive deep models for semantic compositionality over a sentiment treebank." Proceedings of the 2013 conference on empirical methods in natural language processing. 2013.

**Experience Assessment:**

I have published one or two papers in this area.

**Review Assessment: Checking Correctness Of Derivations And Theory:**

I carefully checked the derivations and theory.

**Review Assessment: Checking Correctness Of Experiments:**

I assessed the sensibility of the experiments.

**Review Assessment: Thoroughness In Paper Reading:**

I read the paper thoroughly.

---

> ### Author Response · Authors · 2019-11-15
> **Answer to AnonReviewer3**
>
> We thank AnonReviewer3 for articles [1,2,3]. Though our framework is different, connections with these refs are worthy of attention and we now cite these papers in the introduction.
>
> • “In order to show that the EVT indeed helps empirically in the way that an adversarial classifier enforces the inf-norm of vectors follow the Generalised Extreme Value (GEV) [...] supporting evidence.”
> ➜    Selecting the logistic distribution is not the central point since we apply a standardization whose purpose is to place ourselves in the framework where the considered tail index is equal to 1.
> “From the perspective of learning representations with structured priors, there exists an interesting work on decomposing vector representations into lengths [...]. It would be interesting to see if the proposed method is indeed better than the way that structured priors are enforced in [2].”
> ➜ We thank  AnonReviewer3 for mentioning article [2]. In the newest version  we mention that future work will implement a comparison with this work which we have not done yet due to time constraints.
>
> • “Linguistically, given the distributional hypothesis, the length of learnt vectors tends to be highly correlated with the frequency information of available concepts and the direction of them matters more. [...] would be applicable in fine-grained sentiment analysis, such as Stanford Sentiment Treebank [3].”
> ➜ We make no claim that the proposed method would be applicable in fine-grained sentiment analysis such as [3]. We would like to mention that the extreme values in [3] correspond to annotation labels with sharp and strong sentiment. Such extremes are not the same as the extreme embeddings that we thoroughly study in this paper.
>
> • “The construction of the two datasets seems to be very arbitrary given that there exists a large number of sentiment analysis datasets and many with lots of samples, I am not sure that the results on the chosen constructed two datasets are sufficient enough to support the claim.”
> ➜ The datasets are commonly used datasets for binary classification of text data (see [5, 6]). What are the datasets that  AnonReviewer3 seems to have in mind?
>
> • “The size of the datasets is too small. Given that, the marginal improvement against the NN baseline could be a result of a specific initialisation, which doesn't generalise to other random initialisations.”
> ➜ The mentioned datasets are commonly used datasets for binary classification of text data. We have tried different initializations and obtained similar results. We do not report each initialization in this paper. Concerning the sizes of datasets, we work with a limited amount of GPU time and we cannot go upscale in our experiments as R2 suggests. We want to raise that we have more than 200 extreme samples thus the improvement on the extreme samples is far from marginal. Note that the embeddings are not learnt from scratch: they are built to perform a classification task on top of (frozen) BERT embeddings, please refer to the GLUE benchmark (https://gluebenchmark.com/) for similar system.
>
> • “The dimension of vector representations is also too small. [...] it is in a very high dimensional space.”
> ➜ In the original BERT paper, the size of the embedding used is 768. The size of the learnt embedding in the present work  is a hyperparameter which varies from 10 to 768.  the value 50 was automatically chosen by cross-validation. We now mention this fact in the additional experiment settings for real data section in the Appendix.
>
> • “There are many straightforward distributions [...] a prior on the norm of high dimensional vectors.”
> ➜ As mentioned earlier in our response, In this paper, we do not use an explicit prior on the radius; Instead our target (=prior) is a multivariate extreme value distribution called the Logistic distribution in the EVT setting. It happens that the radial component of this distribution is heavy tailed but this constraint does not need to appear (and does not) in our algorithm.
>
> ---
> [1] Siffer, Alban, et al. "Anomaly detection in streams with extreme value theory." Proceedings of the 23rd ACM SIGKDD International Conference on Knowledge Discovery and Data Mining. ACM, 2017.
> [2] Guu, Kelvin, et al. "Generating sentences by editing prototypes." Transactions of the Association for Computational Linguistics 6 (2018): 437-450.
> [3] Socher, Richard, et al. "Recursive deep models for semantic compositionality over a sentiment treebank." Proceedings of the 2013 conference on empirical methods in natural language processing. 2013.
> [4] Hamid Jalalzai, Stephan Clémencon, and Anne Sabourin. On binary classification in extreme regions. In Advances in Neural Information Processing Systems, pp. 3092–3100, 2018.
> [5] Goodfellow, I., Bengio, Y., & Courville, A. (2016). Deep learning. MIT press.
> [6] R Stewart, S Ermon, Label-free supervision of neural networks with physics and domain knowledge, In Thirty-First AAAI Conference on Artificial Intelligence, 2017.

---

### Official Review · AnonReviewer1 · 2019-10-23
**Official Blind Review #1**

**Rating:** 1

**Review:**

The paper explores learning dilation-invariant sentence representations, with a goal of improving downstream task performance on rare events. A pre-trained embedding is encoded as a latent variable Z, which is constrained to be multi-variate heavy tailed. Separate classifiers are trained on the head and tail of the distribution. Similarly, separate sentence generators are trained on the head and tail of the distribution, in order to allow data augmentation (creating diversity in the outputs by scaling the representation). While the high level motivation and algorithm is interesting, I found the paper very hard to follow, and the experiments are weak.

I have quite a few concerns:
- The algorithm takes a sentence embedding from BERT as input. BERT produces contextualized word representations, not sentence embeddings, so I don't know what the authors did here (the intro also claims that ELMo and GPT learn sentence embeddings, which is also confusing).
- The paper argues that with empirical risk minimization, "nothing guarantees that such classifiers perform satisfactorily
on the tails of the explanatory variables". However, I could not follow what such guarantees the proposed method offers, if any.
- Experiment 4.1 is impossible to follow without reading the appendix. This section should be expanded, or completely moved to the appendix.
- The authors claim without evidence that a baseline of a neural network trained on top of the "BERT embedding" is state-of-the-art for sentiment classification. While there isn't enough information to know what was done, most state-of-the-art approaches involve fine-tuning BERT.
- No comparisons are made with any other work, despite the method attempting a very general and well studied problem of text classification.
- The submission claims that "Applying a dilation is equivalent to assess the generalization of classifiers outside
the envelope of both training and testing samples.". It isn't obvious to me that dilation captures the variation in embeddings you'd get from out-of-domain training samples.
- The authors compare their data augmentation results to "backtranslation". The citation for the method appears to be a class project, and in fact does round-trip translation for paraphrasing, and not back translation.
- No attempt is made to show if the data augmentation approach actually improves end task performance.

**Experience Assessment:**

I have read many papers in this area.

**Review Assessment: Checking Correctness Of Derivations And Theory:**

I did not assess the derivations or theory.

**Review Assessment: Checking Correctness Of Experiments:**

I carefully checked the experiments.

**Review Assessment: Thoroughness In Paper Reading:**

I read the paper at least twice and used my best judgement in assessing the paper.

---

> ### Author Response · Authors · 2019-11-15
> **Answer to AnonReviewer1**
>
>
> • “The algorithm takes a sentence embedding from BERT as input. BERT produces contextualized word representations, not sentence embeddings, so I don't know what the authors did here (the intro also claims that ELMo and GPT learn sentence embeddings, which is also confusing).”
> ➜ AnonReviewer1 is right. BERT produces contextualized word representation which can be applied to sentences (refer to the original paper). In our implementation we use the [CLS] token as an embedding of the full sentence as done in the original paper on the glue benchmark for classification task. We applied BERT on the sentences of the studied dataset as input for the algorithms we detail.
>
> • “The paper argues that with empirical risk minimization, "nothing guarantees that such classifiers perform satisfactorily on the tails of the explanatory variables". However, I could not follow what such guarantees the proposed method offers, if any.”
> ➜ Paper [1] precisely details why the tails deserve a specific treatment. The mentioned paper also provides theoretical guarantees (theorem 2). As advised by R2, we will explicitly state the results from [1] that are relevant for the present paper.
>
> • “Experiment 4.1 is impossible to follow without reading the appendix. This section should be expanded, or completely moved to the appendix."
> ➜ Experiment 4.1 has been moved to the Appendix.
>  The authors claim without evidence that a baseline of a neural network trained on top of the "BERT embedding" is state-of-the-art for sentiment classification. While there isn't enough information to know what was done, most state-of-the-art approaches involve fine-tuning BERT.”
>
> • “No comparisons are made with any other work, despite the method attempting a very general and well-studied problem of text classification.”
> ➜ Our aim is to show that learning a regularly varying representation on top of a baseline representation (such as BERT) improves the classification performance of a standard classifier (such as MLP) compared to applying the same  standard classification algorithm (MLP) to the baseline representation. Table 1 from the new version of the paper precisely gathers the experimental results with regards to this claim.
> Our choice of BERT+MLP as baseline was merely guided by the state of the art approaches when we started working on this project.
>
> • “ The submission claims that "Applying a dilation is equivalent to assess the generalization of classifiers outside the envelope of both training and testing samples.". It isn't obvious to me that dilation captures the variation in embeddings you'd get from out-of-domain training samples.”
> ➜ This sentence has been removed from the newest version for the sake of clarity.
>
> • “The authors compare their data augmentation results to "backtranslation". The citation for the method appears to be a class project, and in fact does round-trip translation for paraphrasing, and not back translation.”
> ➜ The author of the [2] used the word “backtranslation” along their article. We will modify our paper and replace “backtranslation” with “round-trip translation”.
>
> • “No attempt is made to show if the data augmentation approach actually improves end task performance.”
> ➜ Please refer to the last experiment:  we observe that Hydra outperforms all other methods in terms of distinct 1 and distinct 2. Table 1b shows that improvement in F1 score induced by dataset augmentation by Hydra beats all other methods and is only equaled by EDA.
>
> ---
> [1] Hamid Jalalzai, Stephan Clémencon, and Anne Sabourin. On binary classification in extreme regions. In Advances in Neural Information Processing Systems, pp. 3092–3100, 2018.
> [2] Shleifer,S. “Low resource text classification with ulmfit and backtranslation”, arXiv preprint arXiv:1903.09244, 2019.

---

### Official Review · AnonReviewer2 · 2019-10-24
**Official Blind Review #2**

**Rating:** 3

**Review:**

This paper proposes a new embedding method for sentences that aims to preserve dilation invariance.  Much of the methodology is justified by results for extremal point classification under particular assumptions, and then the authors try and encourage these assumptions to be met via penalty terms introduced in their embeddings/augmentation models.  However, while the proposed methodology seems interesting/novel, it remains conceptually unclear why it should be superior to standard text classification methods (ie. exactly what assumptions are being exploited to improve performance and how exactly do those assumptions help should be made more explicit). In particular, why is dilation invariance even a good idea?

Overall, I find the paper a bit mathematically dense in Secs 2.1-2.2, which would not be a bad thing if the math were necessary to justify why the proposed methodology works well, but it in this case seems mainly presented as background material (as if it were a prerequisite to understand the method itself, which it is certainly not).

Why not instead present an explicit theorem providing some statistical guarantees for the proposed methodology in Sec 3.1 based on the constant-along-rays result (which would be nice to have regardless), and then follow up the theorem with background math from 2.1-2.2 which is necessary to understand the proof?

As it is currently written the paper is a bit too dense in terminology, and opaque names like Hydra and Orthrus used to describe straightforward concepts that are essentially a neural classifier (of a particular form) and a seq2seq-based data augmentation procedure (which would be good to describe in language more familiar to the ML audience). In particular, the goals of Hydra and Orthrus should first be intuitively described before delving into their various components.

- Why did the authors never evaluate the overall sentiment prediction performance of Orthrus + Hydra used together vs other classifiers + data augmentation strategies?

- If the goal of dilation invariance is to help the classifiers better generalize to out-of-distribution test sentences, then why not verify this happened, eg. by training on Yelp and testing on Amazon?

- The authors should better justify the assumption of Jalalzai et al, and why this is appropriate for the MLP classifier used later in the paper.

- This statement needs to be clarified and have citation: "Such classifier whose output solely depends on the angle Θ(x) of the considered input, with provable guarantees concerning the classification risk in out-of-sample regions scaling as the square root of the number of extreme points used at the training step"

- The authors should explain Equation (1) in English rather than referring to it so early on the paper (pg 1: "satisfying Equation 1"). I had no idea what this was supposed to mean as a first time reader.

- The way figure 4 is presented is a bit opaque and took me a while to understand (have to look closely at Fig 4a to see the columns are not monochromatic).

- "We also compare Hydra to a Vanilla Sequence to Sequence to demonstrate the validity of our approach" How "Vanilla Sequence to Sequence" (word 'model' is missing) is used for dataset augmentation needs to be clarified here.

- A figure demonstrating an example of the phenomenon explained in Sec 2.2 would be  helpful to aid reader's intuition.

- Typo: "eugmentation"

**Experience Assessment:**

I have published one or two papers in this area.

**Review Assessment: Checking Correctness Of Derivations And Theory:**

I assessed the sensibility of the derivations and theory.

**Review Assessment: Checking Correctness Of Experiments:**

I assessed the sensibility of the experiments.

**Review Assessment: Thoroughness In Paper Reading:**

I read the paper at least twice and used my best judgement in assessing the paper.

---

> ### Author Response · Authors · 2019-11-15
> **Answer to AnonReviewer2**
>
> We thank AnonReviewer2 for spotting the typo.
>
> • “In particular, why is dilation invariance even a good idea?”
> ➜ The dilation invariance is a label invariance of the embeddings. Such invariance, provided by the approach detailed in the paper, allows generating new text data based on labeled inputs while preserving the same label. To the best of our knowledge, no other embedding provides a framework to generate new text data with a label preserving approach.
>
> • “Why not instead present an explicit theorem providing some statistical guarantees for the proposed methodology in Sec 3.1 based on the constant-along-rays result (which would be nice to have regardless), and then follow up the theorem with background math from 2.1-2.2 which is necessary to understand the proof?”
> ➜  Following your suggestion we have added an explicit statement of the statistical guarantees that can be obtained with such constant-along-rays embeddings, citing theorem 1 of [1] in Section 2.1.
>
> • “Why did the authors never evaluate the overall sentiment prediction performance of Orthrus + Hydra used together vs other classifiers + data augmentation strategies?”
> ➜ We point out that the performance of Orthrus is compared to a MLP classifier on similar input (see Table 1 of the new version of the paper). Hydra relies on the regularly varying representation provided by Orthrus. Therefore, the performance of Hydra + Orthrus is compared to state-of-the-art methods in terms of data generation (see Table 2a and Table 2b).
>
> • “If the goal of dilation invariance is to help the classifiers better generalize to out-of-distribution test sentences, then why not verify this happened, eg. by training on Yelp and testing on Amazon?”
> ➜ Although there are works addressing learning a task on a given dataset and testing on a different dataset which relates to transfer learning, we make no claim that the regularly varying embedding allows to do this. The added value of our approach is that it allows:
> 1. Generalization for points which lie out of the envelope of the training inputs,
> 2. A label preserving (text) data augmentation.
>
> • “The authors should better justify the assumption of Jalalzai et al, and why this is appropriate for the MLP classifier used later in the paper.”
> ➜ Following your suggestion we now provide some intuition about the regular variation assumption of Jalalzai et al. in Section 1 and Section 2. In Section 3, we emphasize  that in the present work we do not assume that the original representation (from BERT) satisfies this assumption. Instead we construct a new representation that does, via the GAN machinery with a target with the desired property. Also we have added a few sentences concerning the advantage of using this representation for text document classification, namely the probability ratio between the two classes, conditionally on the input x, solely depend on the angle theta(x) above large radial thresholds, which allows to classify the most extreme test  points using information from a given fraction of the training set corresponding to inputs with largest norm.
>
> • “This statement needs to be clarified and have citation: “Such classifier whose output solely depends on the angle Θ(x) of the considered input, with provable guarantees concerning the classification risk in out-of-sample regions scaling as the square root of the number of extreme points used at the training step”
> ➜ The information that helps to understand this statement is provided in Theorem 2 from [1]. But, indeed, it is not clear as it was explained and we clarified the paper accordingly, by adding the aforementioned theorem in Section 2.2.
>
> • “The authors should explain Equation (1) in English rather than referring to it so early on the paper (pg 1: "satisfying Equation 1"). I had no idea what this was supposed to mean as a first time reader.”
> ➜ We have updated the paper accordingly and explain Equation (1) in the introduction as a homogeneity property above large radial thresholds.
>
> • “A figure demonstrating an example of the phenomenon explained in Sec 2.2 would be helpful to aid reader's intuition.”
> ➜ We have added  a figure (Figure 1) to illustrate the angular classifier in Sec 2.2.
>
> ---
> [1] Hamid Jalalzai, Stephan Clémencon, and Anne Sabourin. On binary classification in extreme regions. In Advances in Neural Information Processing Systems, pp. 3092–3100, 2018.

---

### Decision · Program_Chairs · 2019-12-19

**Decision:**

Reject

**Comment:**

Three reviewers recommend rejection. After a good rebuttal, the first reviewer is more positive about the paper yet still feels the paper is not ready for publication. The authors are encouraged to strengthen their work and resubmit to a future venue.